# Evaluation of Anti-Oxinflammatory and ACE-Inhibitory Properties of Protein Hydrolysates Obtained from Edible Non-Mulberry Silkworm Pupae *(Antheraea assama* and *Philosomia ricinii*)

**DOI:** 10.3390/nu15041035

**Published:** 2023-02-19

**Authors:** Preeti Sarkar, Alessandra Pecorelli, Brittany Woodby, Erika Pambianchi, Francesca Ferrara, Raj Kumar Duary, Giuseppe Valacchi

**Affiliations:** 1Department of Food Engineering and Technology, School of Engineering, Tezpur University, Napaam 784028, Assam, India; 2Department of Environmental and Prevention Sciences, University of Ferrara, 44121 Ferrara, Italy; 3Plants for Human Health Institute, NC Research Campus, NC State University, Kannapolis, NC 28081, USA; 4Department of Chemical, Pharmaceutical and Agricultural Sciences, University of Ferrara, 44121 Ferrara, Italy; 5Department of Dairy Science & Food Technology, Institute of Agricultural Sciences, Banaras Hindu University (BHU), Varanasi 221005, Uttar Pradesh, India; 6Department of Food and Nutrition, Kyung Hee University, Seoul 02447, Republic of Korea

**Keywords:** non-mulberry silkworm pupae, angiotensin converting enzyme-inhibitory peptides, antioxidative peptides, ultrafiltration, anti-inflammatory peptides

## Abstract

Food-derived bioactive peptides (BAPs) obtained from edible insect-protein hold multiple activities promising the potential to target complex pathological mechanisms responsible for chronic health conditions such as hypertension development. In this study, enzymatic protein hydrolysates from non-mulberry edible silkworm *Antheraea assama* (Muga) and *Philosomia ricini* (Eri) pupae, specifically Alcalase (*A. assama)* and Papain (*P. ricini*) hydrolysates obtained after 60 and 240 min, exhibited the highest ACE-inhibitory and antioxidant properties. The hydrolysates’ fractions (<3, 3–10 and >10 kDa), specifically Alc_M60min_F3 (≤3 kDa) and Pap_E240min_F3 (≤3 kDa), showed the highest antioxidant and ACE-inhibitory activities, respectively. Further RP-HPLC purified sub-fractions F4 and F6 showed the highest ACE inhibition as well as potent anti-oxinflammatory activities in lipopolysaccharide (LPS)-treated endothelial cells. Indeed, F4 and F6 ACE-inhibitory peptide fractions were effective in preventing p65 nuclear translocation after 3 h of LPS stimulation along with the inhibition of p38 MAPK phosphorylation in HUVEC cells. In addition, pretreatment with F4 and F6 ACE-inhibitory peptide fractions significantly prevented the LPS-induced upregulation of COX-2 expression and IL-1β secretion, while the expression of NRF2 (nuclear factor erythroid 2-related factor 2)-regulated enzymes such as HO-1 and NQO1 was induced by both peptide fractions. The derived peptides from edible pupae protein hydrolysates have potentialities to be explored as nutritional approaches against hypertension and related cardiovascular diseases.

## 1. Introduction

Hyperactive or dysregulated renin-angiotensin systems and oxinflammatory processes are the primary key players, among many, participating in the pathology of hypertension [1,2,3]. Angiotensin II generation promotes long-term unfavorable results on vascular health via oxidative stress, decreased nitric oxide, pro-inflammatory cytokines production and vascular remodeling [2]. Thus, endothelial dysfunction caused by hypertension, vascular inflammation and arterial remodeling finally contributes to cardiovascular disease progression [4]. Due to severe side effects associated with classical therapies utilized in the treatment of cardiovascular disorders [5], there is a huge call to find alternative natural sources. For the last two decades, extensive research work has been carried out on protein-derived bioactive peptides (BAPs) as valuable ingredients of functional foods and/or nutraceuticals. Due to their abundance in any proteinaceous material, multifunctional properties, extreme stability, safety and bioavailability, they have been appreciated by researchers as next-generation potential therapeutics to promote health and prevent the risk of chronic diseases [6].

Several food products made from insects’ flour are nowadays available commercially due to the efficient functional properties present in their proteins. However, research on the use of insect-protein-extracted peptides as potential therapeutics is relatively new and trending. Among different species of silkworms found around the world, *Bombyx mori* pupae have long been commonly used as food and therapeutics in countries such as Korea, Japan, Thailand, China and India [7,8]. However, much less studies have been carried out for the other members of the silkworm. Two non-mulberry silkworm species, *Antheraea assama* and *Philosomia ricinii* belonging to the Saturniidae family, are predominantly found in the state of Assam and in a few other regions of northeastern India. These silkworms are abundantly used for rearing their golden and white silk threads, locally known as Assam silk or “Muga and Eri silk” [9]. As a result, in the local reeling industries, a large quantity of pupae waste accumulates, which is either regarded as agro-industrial waste, arising environmental problems, or commercially used by the local communities as a delicacy in its cooked and fermented form [9]. Since insects have a notable protein content ranging from 21 to 80% depending on their species and life cycle stage [10], they are therefore enriched with a wide range of bioactive peptides with yet undisclosed potentialities.

Until now, no studies have been carried out on the natively available non-mulberry edible silkworm pupae obtained as indigenous reeling industrial waste to evaluate the presence of multifunctional peptides. Therefore, this is a new investigation on Muga and Eri pupae-derived whole protein extracts for the identification of ACE-inhibitory and anti-oxinflammatory peptides with the aid of six different enzyme applications. Consequently, exploring the possibility of generating bioactive peptides or hydrolysates from these unconventional edible insects could be an interesting avenue of research to facilitate the development of pharmaceutical or nutraceutical products with the potential to target hypertension and oxinflammatory phenomena, and, thus, to manage cardiovascular diseases.

## 2. Materials and Methods

### 2.1. Materials and Chemicals

*Philosomia ricinii* (Local name: Eri) and *Antheraea assama* (Local name: Muga) pupae were collected from a sericulture farm located in the Sonitpur district, and some pupae were also procured in the month of March, 2017, from the local market situated in Nelie, Nagaon district, Assam, India. Samples were immediately kept at 25 °C inside zip locked plastic pouches and taken to the laboratory. Pupae were taxonomically identified and confirmed by the Zoological Survey of India in Kolkata, India. Pupae were manually cleaned to remove dirt and other impurities and then stored at −20 °C, until further experimental studies. Pupae were freeze-dried (Lyolab, Lyophilization Systems Pvt. Ltd., Hyderabad, India), ground and stored at 4 °C until utilized. Pupae powder was mixed with n-hexane in a ratio of 1:5 (*w*/*v*), and thus defatted twice in Soxhlet apparatus (6 h) followed by oven drying (NDO 700W, Eyela, Singapore) at 50 °C.

All enzymes, including Alcalase (EC 3.4.21.14, from *Bacillus licheniformis*, ≥0.75 Anson U mL^−1^), Flavourzyme (EC 232-752-2, from *Aspergillus oryzae*, ≥500 U g^−1^), Thermolysin (EC 3.4.24.27, from *Geobacillus stearothermophilus,* 30–175 U mg^−1^), Papain (EC 3.4.22.2, from *Carica papaya*, ≥30,000 USP-U mg^−1^), Pepsin (EC 3.4.23.1, from porcine gastric mucosa, 3200–4500 U mg^−1^), Trypsin (EC 3.4.21.4, from porcine pancreas, 1000–2000 BAEE U mg^−1^) and ACE (angiotensin converting enzyme) (EC 3.4.15.1, from rabbit lung, ≥2.0 U mg^−1^) were purchased from Sigma–Aldrich Co. (St. Louis, MO, USA). Chemicals such as hippuric acid (HA), hippuryl-L-histidyl-L-leucine (HHL), 3-(4, 5-dimethylthiazol-2-yl)-2, 5-diphenyl tetrazolium bromide (MTT), 1,1-diphenyl-2-picrylhydrazyl (DPPH), 2,2′-azino-bis 3-ethylbenzthiazoline-6- sulphonic acid (ABTS) and 2,4,6-Tris (2-pyridyl)-s-triazine (TPTZ) were also obtained from Sigma–Aldrich Co. (St. Louis, MO, USA). Specific antibodies against heme oxygenase 1 (HO-1, cat. VPA00553; Bio-Rad Laboratories, Inc. Hercules, CA, USA; ), *NAD*(P)H quinone dehydrogenase 1 (NQO1, cat. VMA00039; Bio-Rad Laboratories, Inc. Hercules, CA, USA;), p65 (cat. sc-8008, Santa Cruz Biotechnology, Inc. Santa Cruz, CA, USA), phospho (p)-p38 (cat. sc-7973, Santa Cruz Biotechnology, Inc. Santa Cruz, CA, USA) and p38 (cat. sc-7149, Santa Cruz Biotechnology, Inc. Santa Cruz, CA, USA) were used for immunoblotting. All other chemicals and reagents used were analytical grade.

### 2.2. Extraction of Whole Protein and Preparation of Protein Hydrolysates

Proteins from the defatted pupae powder of both silkworms were isolated using the alkaline extraction method, as described by Jia et al. [11] with modifications. Briefly, 5% (*w*/*v*) defatted powder in water was calibrated to pH 9.5, using 1 M NaOH. The solution was then stirred for 2 h at room temperature and then centrifuged at 4 °C for 30 min at 8000× *g* (F-35-6-30 rotor, Eppendorf 5430R, Eppendorf AG, Hamburg, Germany). The collected pellet was re-extracted twice using the same procedure. Supernatants obtained from the total four extractions were adjusted to pH 4.0–5.8, using 1 M HCl to screen the isoelectric point. The pellet, having precipitated proteins at pH 4.7, was isolated from the supernatant after centrifugation for 30 min and 8000× *g* at 4 °C. The pellets labeled as MPPC (Muga pupae protein concentrate) and EPPC (Eri pupae protein concentrate) were neutralized with 0.2 M NaOH before lyophilization and stored at −20 °C.

MPPC and EPPC were hydrolyzed with Alcalase (50 °C, pH 8.0), Flavourzyme (50 °C, pH 7.0), Thermolysin (70 °C, pH 8.0), Papain (60 °C, pH 6.0), Pepsin (37 °C, pH 1.7) and Trypsin (37 °C, pH 7.5) at 0.5% enzymes: substrate (E/S) ratio. The protein concentration was adjusted to 5.0% in each reaction mixture and the solutions were set to their appropriate temperature and pH point. During the reaction, pH was kept constant using either 1 M NaOH or HCl. Finally, the digestions were performed in a shaking water bath for a total of 6 h. Five hundred µL samples were taken out after 15, 30, 60, 120, 180, 240, 300 and 360 min to measure their degree of hydrolysis (DH) and ACE inhibition. At the reaction end, to inactivate the enzymes, each hydrolysate was boiled for 10 min, cooled and centrifuged at 10,000 × g for 10 min at 20 °C. Finally, the supernatants containing the peptide mix were collected and labeled accordingly. The pupae proteins treated at the same conditions for each protease with no protease addition were used as controls. The degree of hydrolysis (DH) % was calculated using the trinitro benzene sulfonic acid (TNBS) method given by Hall et al. [12].

### 2.3. ACE-Inhibition Activity Determination

ACE-inhibitory activities of hydrolysates were estimated following the protocol described by Hall et al. [13]. The relative amounts of released HA and uncleaved HHL were quantified by HPLC (high pressure liquid chromatography) (Model 600E, Waters Corporation, Milford MA, USA) on an analytical C18 column (YMC Pack ODSAM 12,505–2546 wt, YMC America, Inc., Allentown, PA, USA). ACE inhibition (%) was calculated as follows:$$\mathrm{ACE}\;\mathrm{inhibition}=1-\frac{A_{\mathrm{inhibitor}}}{A_{\mathrm{control}}}\times 100$$
where A_inhibitor_ and A_control_ denote the relative areas (A) of HA peaks in the presence or absence of inhibitors.

### 2.4. Determination of Antioxidant Activities

#### 2.4.1. DPPH Radical Scavenging Assay

The analysis of hydrolysates for the DPPH radical scavenging potential was conducted as per the method illustrated by Hall et al. [13], in a microtiter plate (Thermo Fisher Scientific Inc., Waltham, MA, USA). The degree of DPPH radical scavenging activity was calculated based on the scavenging activity (%). All the results were indicated as the EC_50_ (half maximal effective concentration) value.

#### 2.4.2. ABTS Radical Scavenging Assay

The analysis of hydrolysates for the ABTS radical scavenging potential was conducted according to You and Wu [14], in a microtiter plate. The degree of ABTS radical scavenging activity was evaluated based on the scavenging activity (%) after absorbance was recorded at 1 min intervals for a total of 8 min. All the results were indicated as the EC_50_ value.

### 2.5. Peptide Fractionation Using Ultrafiltration

Muga and Eri pupae protein hydrolysates with the highest potential of in vitro ACE inhibition and antioxidant activities were fractionated using Amicon^®^ Ultra 15 mL centrifugal filters with molecular weight (MW) cutoff membranes of 3 kDa and 10 kDa (EMD Millipore; Billerica, MA, USA). The resultant filtrates of MW < 3 kDa, 3 < MW < 10 KDa and MW > 10 KDa were later freeze-dried to determine their biological activities.

### 2.6. Peptide Fractionation Using RP-HPLC

In total, 100 mg mL^−1^ of ultramembrane filtrates having the strongest ACE inhibition and antioxidant activities was dissolved in distilled water. Furthermore, 100 μL of each sample was loaded into RP-HPLC (Waters Corporation, Milford, MA, USA) connected to a PrepTM C18 column (7.8 mm × 300 mm, Milford, Waters). The elution of the sample was performed in linear gradient mode at a flow rate of 1 mL min^−1^ from 100% eluent A (0.1% trifluoroacetic acid (TFA) in double distilled water) for 2 min; 0 to 25% of eluent B (0.1% TFA in acetonitrile) for 30 min; 25% to 100% at 45 min then back to 0% at 50 min. The eluted fractions were monitored at 214 nm absorbance followed by fraction collection according to each peak. Purification steps were replicated until sufficient samples were accumulated for the ACE-inhibition assay.

### 2.7. Human Umbilical Vein Endothelial Cell Line (HUVECs) Culture

HUVECs were obtained from the American Type Culture Collection (ATCC, Manassas, VA, USA). The cells were cultured in endothelial growth medium (EGM, Gibco, Burlington, ON, Canada) with 50 units mL^−1^ penicillin-streptomycin (Gibco, Burlington, ON, Canada) and 10% fetal bovine serum (FBS, Hyclone Co., Logan, UT, USA). The cells were maintained at 37 °C, in a 5% CO_2_ atmosphere, at 95% humidity in a cell culture incubator.

### 2.8. Cell Viability Assay

The effects of peptide fractions on endothelial cell viability were determined using the MTT assay. Briefly, HUVECs (1 × 10^5^ cells mL^−1^) were seeded in 96-well plates. Cells were then treated with peptide fractions at concentrations of 0, 10, 20, 50, 100, 300, 500 and 1000 µg mL^−1^ for 24 h. At the end of the incubation period, 0.5 mg mL^−1^ MTT solution was added into each well and incubated for 4 h further. Later, the culture medium was removed, and dimethyl sulfoxide (150 µL well^−1^) was added to each well to solubilize the formazan crystals with thorough shaking for 15 min. The absorbance was read at 570 nm in a SYNERGY H1 microplate reader (BioTek Instruments, Inc., Winooski, VT, USA).

### 2.9. Immunocytochemistry

HUVECs were cultured on cover slips at 1 × 10^5^ cells mL^−1^ density. HUVECs were pretreated with peptide fractions (20 µg mL^−1^) for 24 h and then stimulated with 1 µg mL^−1^ of LPS for 3 h. After LPS treatment, cells were fixed in 4% paraformaldehyde in PBS (phosphate buffer saline) at 4 °C for 30 min, and then permeabilized with 0.25% Triton X-100 in PBS. Blocking was performed at room temperature for 1 h using PBS containing 1% BSA (bovine serum albumin). Cover slips were then incubated with the primary antibody for nuclear factor kappa-light-chain-enhancer of activated B cells (NF-κB) p65 subunit (cat. sc-372, Santa Cruz Biotechnology, Inc. Santa Cruz, CA, USA), diluted 1:400 in PBS including 0.25% BSA overnight at 4 °C. The next day, the cells were incubated for 1 h with fluorochrome-conjugated secondary antibody Alexa Fluor 488 (Invitrogen A11008, Thermo Fisher Scientific Inc., Waltham, MA, USA), diluted 1:200 in PBS including 0.25% BSA. After removal of the secondary antibody, nuclei were stained with 1 µg mL^−1^ diamidino 2-phenylindole (DAPI) (Invitrogen D1306, Thermo Fisher Scientific Inc., Waltham, MA, USA) for 1 min. Using the PermaFluor mounting medium (Thermo Fisher Scientific Inc., Waltham, MA, USA), cover slips were mounted onto glass slides. Images were acquired using a Zeiss LSM10 confocal microscope at 40× magnification and analyzed using ImageJ software (version 1.53c 26 June 2020; National Institutes of Health, Bethesda, MD, USA) [15].

### 2.10. Protein Extraction and Western Blot Analysis

Cells were seeded at a density of 5 × 10^6^ cells mL^−1^ in a 48-well plate and pretreated with peptide fractions (20 µg mL^−1^) for 24 h and then treated with 1 µg mL^−1^ LPS for different time points. After treatment, cells were detached and washed with 1× ice-cold PBS, and cell pellets were lysed with ice-cold RIPA (radioimmunoprecipitation assay) buffer (cat. AAJ62524AD, Alfa Aesar, Tewksbury, MA, USA) containing 1% protease (cat. 78430, Thermo Fisher Scientific Inc., Waltham, MA, USA) and 1% phosphatase (cat. 1862495, Thermo Fisher Scientific Inc., Waltham, MA, USA) inhibitor cocktails. Equivalent amounts of protein were loaded on 8% sodium dodecyl sulphate polyacrylamide gel electrophoresis (SDS-PAGE) gels, and then transferred to nitrocellulose membranes. Immunoblotting was conducted with targeted primary antibodies and peroxidase-conjugated anti-mouse and anti-rabbit secondary antibodies (cat. 170-6515 and 170-6516; Bio-Rad Laboratories, Inc. Hercules, CA, USA) dil. 1:10 000. Protein bands were detected by using the Clarity Western ECL Substrate Kit (cat. 1705060, Bio-Rad Laboratories, Inc., Hercules, CA, USA) and ChemiDoc MP Imaging System hardware and software (Bio-Rad Laboratories, Inc. Hercules, CA, USA). Protein bands were normalized to the loading control β-actin (cat. A3854; Sigma–Aldrich Co., St. Louis, MO, USA). Images of bands were analyzed by using ImageJ software (version 1.53c 26 June 2020; National Institutes of Health, Bethesda, MD, USA) [16].

### 2.11. COX-2 and IL-1β Detection Using ELISA

Briefly, HUVECs were cultured in a 6-well plate at a density of 5 × 10^6^ cells mL^−1^ and pretreated with peptide fractions (20 µg mL^−1^) for 24 h and then treated with 1 µg mL^−1^ LPS for 12 h. Later, the culture medium was collected and stored at ≤ −70 °C. The cells were rinsed two times with PBS and then lysed within the wells by the addition of the Cell Extraction Buffer. After centrifugation at 2000× *g* for 5 min, cell lysates were stored at ≤ −70 °C. IL-1β released in the media and COX-2 cellular levels were determined using specific ELISA kits (cat. E00021, Human IL1-beta ELISA Kit, Proteintech Group, Inc, Rosemont, IL, USA; Catalog #: DYC4198-2, Human/Mouse Total COX-2 DuoSet IC ELISA, R&D Systems, Inc., Minneapolis, MN, USA), as per the manufacturer’s directions. Detection was carried out using Gen5 software (BioTek Instruments, Inc., Winooski, VT, USA).

### 2.12. Statistical Analysis

Every experiment was carried out in at least three independent experiments, and data are indicated as mean ± standard deviation. Results were statistically analyzed, using SPSS 19.0 version (SPSS Inc., Chicago, IL, USA). All the data collected were used for a one-way analysis of variance (ANOVA). *p* < 0.05 significance level was used to correlate the means by employing Duncan’s test.

## 3. Results and Discussion

### 3.1. Effects of Different Proteases on the Degree of Hydrolysis

In this study, we optimized the proteolytic digestions of MPPC and EPPC in the presence of six different proteases along with hydrolysis time (0 to 360 min) to evaluate the possibility of releasing ACE-inhibitory hydrolysates with the highest activity. The effect of the relationship between applied enzymes at the same concentration and the incubation time on the DH of each protein concentrate (MPPC and EPPC) is plotted in Figure 1. In both the cases, protein hydrolysis increased rapidly up to a certain time point of approximately 60 min and then gradually became stable, entering a static phase towards the end of hydrolysis. Slower rates of hydrolysis displayed by all the proteases, except Thermolysin and Pepsin, are likely due to either competitive proteolytic inhibition by the hydrolysis products (peptides or amino acids) or cleavage saturation due to the scarcity of available peptide bonds left to be cleaved. Although both the protein concentrates were reactive to all the enzymes, they showed a different sensitivity. The variation in hydrolysis patterns observed for both the protein concentrates could be due to species-specific differences in protein composition and their amino acid order. Based on our results, we observed that Alcalase was found to be the most effective protease in hydrolyzing both MPPC and EPPC, while Trypsin was the lowest as well as much weaker than Pepsin, likely due to lower amounts of Trypsin cleavage sites. Some authors have previously established the prevalent use of Alcalase for producing insect-protein hydrolysates from crickets, mealworm larvae and *Bombyx mori* pupae due to its broad specificity for cleavage sites [11,13,17]. The relative highest DH for MPPC achieved by each enzyme, at the end of reaction, was in the following order (Figure 1A):

Alcalase (69.49 ± 1.04%) > Thermolysin (37.77 ± 1.33%) > Pepsin (33.22 ± 1.45%) > Flavourzyme (25.16 ± 0.32%) > Papain (23.37 ± 1.54%) > Trypsin (9.38 ± 1.08%).

For EPPC, DH variation ranged from 5.59 to 65.73%. Thus, the relative highest DH of Eri pupae protein obtained by each enzyme after 360 min was the following (Figure 1B):

Alcalase (65.73 ± 0.98%) > Flavourzyme (59.24 ± 1.25%) > Thermolysin (38.77 ± 1.33%) > Pepsin (26.05 ± 0.45%) > Papain (24.38 ± 1.54%) > Trypsin (11.57 ± 1.08%).

We also observed that only the control solution based on Thermolysin parameters achieved a hydrolysis rate of 7.2% for MPPC and 8.0% for EPPC after 360 min of incubation (not shown). This indicates that heating at 70 °C, the optimum temperature for Thermolysin activity, might have played a significant role in enhancing protein cleavage. The DH value seen in this study for Alcalase hydrolysate (Muga and Eri) after 2 h was potentially greater than the published data shown by Dion Poulin et al. [18] for *G. sigillatus* and *T. molitor* (28.1 to 33.8% at 3.0% (E/S)) and Purschke et al. [19] for *Locusta migratoria* (13.3–15.2%; E/S = 1.0%). The results were consistent with data (29.2%–51.2%; at 0.5%, 1.5% and 3.0% E/S ratio) studied by Hall et al. [12,13] after 30–90 min for the same insects at 50% (*w*/*v*) concentration. Such variation might be influenced by prior heat treatment at 90 °C that can affect DH. Compared to DH results achieved for *A. diaperinus* (20.5% (0.5% E/S), 20.3% (1.5% E/S) and 25.0–28.4% (3.0% E/S); 4 h) using Alcalase as presented by Sousa et al. [20], these values were also considerably higher. DH values obtained for Flavourzyme-derived hydrolysates for Muga pupae were found higher than the reported data of cricket protein (33.0%; 2.0% E/S), but the DH noted for Eri pupae was similar to the values presented for mealworm protein (51.0%) at similar conditions [21]. These differences can be explained by various factors such as protein structure, amino acid sequence and other treatment conditions.

### 3.2. ACE-Inhibitory Potential of Hydrolysates

The effects of protease treatment at different time conditions on the ACE-inhibitory potential of Muga and Eri pupae protein-derived hydrolysates are presented in Figure 2, while the correlations between the resultant DH and ACE inhibition are represented in Appendix A. Based on our data, the controls displayed no remarkable inhibition over time for all six enzymes. Even the increased resultant DH during Thermolysin hydrolysis conditions showed no significant effect on the ACE-inhibitory potential. More than 70% of the ACE-inhibition potential of MPPC-derived hydrolysates was obtained after 60 min (74.46 ± 2.44%) for Alcalase; 300 min (71.07 ± 2.05%) and 360 min (73.59 ± 2.39%) for Thermolysin (*p* > 0.05); 180 min (72.89 ± 1.36%) and 240 min (77.35 ± 1.57%) for Pepsin as shown in Figure 2A. For EPPC (shown in Figure 2B), the ACE-inhibition potential >70% was observed after 120 min (74.67 ± 1.70%) and 180 min (77.05 ± 2.12%) for Alcalase (*p* > 0.05); 180 min (81 ± 2.36%) and 240 min (83.21 ± 2.51%) for Papain (*p* > 0.05); 60 min (75.31 ± 2.57%), 120 min (78.80 ± 1.79%) and 180 min (82.07 ± 2.56%) for Pepsin. It was noted that with the extended hydrolysis time, the ACE-inhibition activities exhibited by Alcalase-, Papain- and Pepsin-treated hydrolysates were mostly unstable for both types (Appendix A), despite the moderately increasing DH with time. This was probably because with the continuous cleavage process, the newly formed peptides may have either a strong affinity for the ACE functional active site or may be further cleaved over time, resulting in the production of a large number of free amino acids (FAA) and more or less active peptides, as suggested by Salampessy et al. [22]. Our results are in line with a previous study conducted by Dai et al. [17], who revealed that the highest ACE-inhibitory activity of *T. mollitor* was acquired with 20% DH, and additional digestion resulted in a decline in ACE inhibition. Altogether, Alcalase and Pepsin are commonly efficient at producing ACE-inhibitory peptides with >70% inhibition from both Muga and Eri silkworm pupae-extracted proteins.

Therefore, the hydrolysates that exhibited potent ACE-inhibitory activity >70% were finally selected and evaluated for their minimum inhibitory concentration (IC_50_) against ACE. The IC_50_ values of the selected hydrolysates ranged between 0.12 ± 0.01 and 1.71 ± 0.29 mg mL^−1^, respectively (Table 1). Overall, Alc_M60min (Alcalase-treated Muga pupae protein hydrolysate at 60 min) and Pep_M240min (Pepsin-treated Muga pupae protein hydrolysate at 240 min) exhibited the strongest activity. Additionally, no significant differences were observed within the hydrolysates with IC_50_ values that ranged between 1.29 and 1.71 mg mL^−1^. Compared to earlier studies carried out for insect-derived BAPs, the ACE-inhibition IC_50_ values of hydrolysates obtained in this study with Alc_M60min, Pep_M240min and Pap_E240min (Papain-treated Eri pupae protein hydrolysate at 240 min) were slightly lower than the hydrolysates generated from *Tenebrio molitor* larva, *Spodoptera littoralis* and mulberry silkworm (*B. mori*) [17,23,24], demonstrating the productive ACE-inhibiting capabilities of non-mulberry silkworm pupae protein-derived peptides. However, the lowest IC_50_ values found in our study were higher than those of the ultrasonic-pretreated Alcalase-derived hydrolysates of silkworm pupae protein (91.3 µg mL^−1^) [11]. Such a variation is primarily related to the differences in insect species and processing methods.

### 3.3. Antioxidative Potential of Selected Hydrolysates

Since the proteolytic hydrolysis of total protein extracts can give rise to a number of active peptides in a hydrolysate, it seems reasonable to presume that the hydrolysates generated in our study might display versatile behaviors. Therefore, the antioxidant properties of eight selected hydrolysates (shown in Table 1) were also evaluated.

The electron-donating ability in the tested hydrolysates was investigated using the DPPH and ABTS radical scavenging assays: the primary mechanism by which antioxidants can inhibit oxidative propagation. Tested hydrolysates demonstrated that their EC_50_ values for DPPH radical scavenging activity ranged from 0.48 ± 0.27 to 3.88 ± 0.91 mg mL^−1^, where hydrolysate Pap_E240min was the most effective at scavenging DPPH, and Pep_E60min (Pepsin-treated Eri pupae protein hydrolysate at 60 min) was the least. In contrast to DPPH results, Pep_E60min hydrolysate exhibited the strongest ABTS radical scavenging activity with no significant difference from Pap_E240min hydrolysate (*p* > 0.05), while the lowest value was shown by Pep_M180min (Pepsin-treated Muga pupae protein hydrolysate at 180 min). The EC_50_ values attained by the ABTS method were higher than those found by the DPPH method, which signifies that DPPH was more sensitive towards the scavenging ability of protein hydrolysates at lower sample concentrations.

Based on our data, it can be confirmed that the selected hydrolysates also efficiently displayed antioxidant capacities. However, the strongest antiradical property was lower than the IC_50_ value obtained for the raw (5.3 µg/mL), boiled form of *T. molitor* (28.9 µg/mL), baked *G. sigillatus* and *S. gregaria* (28.5 µg/mL) [25]. These values were also less than the ones obtained for *Zophobas morio* (4.6 µg/mL) against ABTS, and *Amphiacusta annulipes* (19.1 µg/mL) and silkworm protein (57.91 µg/mL) hydrolysate against DPPH achieved after in vitro gastrointestinal digestion [26,27].

### 3.4. Antioxidative and ACE-Inhibition Potential of Ultrafiltrates

When the ACE-inhibitory and antioxidative properties of eight selected hydrolysates were analyzed, the hydrolysates Alc_M60min and Pap_E240min presented the most efficient bioactivities relative to the others. From a pharmacology point of view, molecules with lower molecular weight generally have greater pharmaceutical properties and nutraceutical uses [28]. Therefore, the next step in our study was the separation of bioactive peptide fragments from Alc_M60min and Pap_E240min hydrolysates, based on their molecular sizes using membrane ultrafiltration. Therefore, these two hydrolysates were further fractionated into different molecular weight cutoffs: MW < 3 kDa, 3 < MW < 10 KDa and MW > 10 KDa. Among them, fraction three of Alc_M60min (IC_50_, 0.49 ± 0.25 µg mL^−1^) and Pap_E240min (IC_50_, 0.67 ± 0.16 µg mL^−1^) hydrolysates had significantly higher binding abilities to ACE (*p* < 0.05) (Table 2). However, all the fractions demonstrated different antioxidant capabilities (Table 2) depending on the substrate, among which the third and second fractions of the Alc_M60min hydrolysate demonstrated, respectively, the strongest DPPH (EC_50_, 90.21± 3.27 µg mL^−1^) and ABTS (EC_50_, 56.35 ± 3.11 µg mL^−1^) free radical scavenging ability. Otherwise, fraction three of the Pap_E240min hydrolysate held a higher DPPH (EC_50_, 49.32 ± 4.09 µg mL^−1^) and ABTS scavenging ability (EC_50_, 19.21 ± 3.64 µg mL^−1^). Altogether after filtration, the fractions < 3 kDa showed significantly higher ACE-inhibitory and antioxidative potentials. The highest antioxidant potential of <3 kDa peptide fraction (from Eri pupae) had a lower EC_50_ value than the studied fractions obtained from *T. molitor* (24.31 µg/mL) [25].

### 3.5. Fractionation of Peptides in Alc_M60min_F3 and Pap_E240min_F3

After fractionation through ultrafiltration membranes, the peptides present in the Alc_M60min and Pap_E240min hydrolysates were segregated into specified sizes and with discrete hydrophilic and hydrophobic proportions. To achieve ACE-inhibitory peptides, the most potent ultrafiltrate fractions, Alc_M60min_F3 (Figure 3A) and Pap_E240min_F3 (Figure 3B), were again separated based on their hydrophobicity by RP-HPLC. Chromatography profiles revealed the complexity of the peptide mixture due to their hydrophilic/hydrophobic composition. Various peaks with inhibitory potential were obtained. Based on their highest activity, two major peptides’ sub-fractions termed F4 (0.07 µg mL^−1^) and F6 (0.03 µg mL^−1^) acquired from Alc_M60min_F3 and Pap_E240min_F3, respectively, were preferred. Specified fractions were accumulated, lyophilized and then used in in vitro experiments to evaluate their activities.

### 3.6. Anti-Oxinflammatory Activity of Purified Peptide Fractions F4 and F6 in Human Endothelial Cells

Inflammation is one of the key participants in the progression of hypertension and its downstream effects. Furthermore, inflammatory processes in hypertension are strongly associated with oxidative stress events, generating a vicious circle of oxinflammation [1,2,3,4]. Therefore, the next step of our study was to understand if the ACE-inhibitory activity of purified peptide fractions F4 and F6 could affect endothelial oxinflammation. Understanding the activity of food-derived bioactive peptides on the pathophysiology of hypertension could allow researchers to characterize and design more potent peptides to manage hypertension and associated cardiovascular diseases.

To characterize the effects of peptide fractions on cell viability, HUVECs were treated for 24 h with different doses of F4 and F6, ranging from 10 µg mL^−1^ to 1000 µg mL^−1^. As indicated in Figure 4, it was noticed that high doses of F4 and F6, above 100 µg mL^−1^, reduced cellular viability below 70%. Thus, for the subsequent experiments, 20 µg mL^−1^ of each peptide fraction was chosen to evaluate the anti-oxinflammatory activities in endothelial cells.

To assess their anti-inflammatory potential, first, we analyzed whether F4 and F6 peptide fractions can affect the nuclear translocation of the major proinflammatory transcription factor NF-κB (nuclear factor kappa-light-chain-enhancer of activated B cells). As shown in Figure 5A, 1 µg mL^−1^ LPS stimulation for 3 h promoted significant p65 nuclear translocation, whereas pretreatment with F6 for 24 h was significantly effective in preventing this effect. However, F4 did not significantly decrease LPS-induced NF-κB activation, although a slightly decreasing trend was evident. Our results suggest that the hydrolysate fractions can promote anti-inflammatory effects due to their ability to regulate NF-κB. In fact, we observed that treatment with the hydrolysate fractions could suppress the LPS-induced phosphorylation of p38 MAPK (Figure 5B), a critical event in MAPK signaling, which can induce NF-κB activation. Whether the hydrolysate fractions can suppress the activation of other MAPK signaling pathways, such as ERK and JNK, needs to be further assessed.

To further confirm whether the silkworm F4 and F6 peptide fractions could suppress inflammation, we evaluated the levels of cyclooxygenase-2 (COX-2), which is an enzyme responsible for the production of the proinflammatory mediators such as prostaglandins [29]. As shown in Figure 5C, COX-2 expression markedly increased by 10.1-fold in LPS-treated cells, compared with untreated cells. Furthermore, pretreatment with F4 and F6 reduced the COX-2 levels by 6.03% and 62.83%, respectively, in contrast to LPS-treated cells. Thus, F6 was more effective in preventing an LPS-induced increase in COX-2 levels.

Since IL-1β is a proinflammatory cytokine involved in hypertension and cardiovascular disease [30], we also examined whether treatment with F4 and F6 peptide fractions could affect IL-1β secretion. We observed that treatment with both peptide fractions, particularly F4, significantly suppressed the LPS-induced increase in IL-1β secretion (Figure 5D). Thus, our results demonstrated that non-mulberry silkworm pupae-purified peptide fractions F4 and F6 suppress LPS-induced cellular inflammatory responses in HUVECs due to their ability to regulate NF-κB nuclear translocation via p38 MAPK signaling, and, consequently, the expression and release of proinflammatory mediators such as COX-2 and IL-1β, respectively.

Due to the antioxidant activity shown by the <3KDa fractions of both F4 and F6, as a next step in the evaluation of their potential ability to affect oxinflammatory processes, we analyzed levels of the antioxidant enzymes heme oxygenase-1 (HO-1) and NAD(P)H quinone dehydrogenase 1 (NQO1) in LPS-treated endothelial cells. LPS treatment alone did not affect HO-1 expression levels, but slightly increased NQO1 expression compared to control cells (Figure 6A,B). However, we observed increased levels of both HO-1 and NQO1 in cells treated with the peptide fractions alone and in combination with LPS after 18 h of treatment. The increased expression of the two antioxidant enzymes after the treatment of HUVECs with F4 and F6, both in the presence and absence of LPS, could imply a potential antioxidant activity of both peptides, which probably occurs through the activation of the NRF2 pathway [31]. NRF2 activation via pharmacological treatment is a potential therapeutic perspective for several chronic diseases such as cardiovascular, neurodegenerative and metabolic burden characterized by oxidative stress and inflammation [32]. Thus, the increased expression of these enzymes further improves the anti-oxinflammatory potential of non-mulberry silkworm pupae-purified peptide fractions. However, further studies are needed to determine how F4 and F6 are involved in the mechanisms underlying NRF2-regulated signal transduction, and to better unravel their effects on the crosstalk between Nrf2 and NF-κB [33].

## 4. Conclusions

This study illustrates the efficient retrieval of underexploited protein from non-mulberry silkworm waste generated after the reeling process, along with the production of its hydrolysates, which displayed interesting bioactivities. We have demonstrated that enzyme-specific hydrolysis is a very important step in validating the production of multifunctional peptides that could have potential benefits for human health. In fact, our results indicate a possible correlation between the ACE-inhibitory and antioxidant properties of <3KDa peptide fractions, observed in vitro, and their potential anti-oxinflammatory activities, validated in LPS-treated endothelial cells. Future research is required to isolate and characterize potent peptides for each bioactivity and to evaluate the inhibition modes of ACE, the mechanisms involved in the anti-oxinflammatory activity and the efficacy and bioavailability of these peptides in vivo. Overall, our findings provide initial data to support the therapeutic potential of using non-mulberry silkworm pupae proteins as a source to generate multifunctional hydrolysates, which can be incorporated into pharmaceutical products such as functional foods and considered as natural alternatives to treat hypertension and related cardiovascular diseases.

## Figures and Tables

**Figure 1 nutrients-15-01035-f001:**
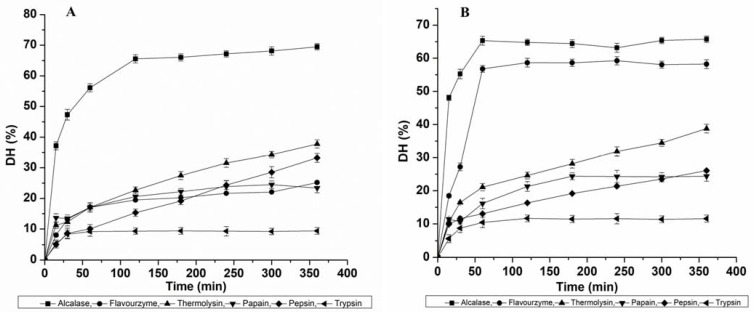
Effect of six proteases (Alcalase, Flavourzyme, Thermolysin, Papain, Pepsin and Trypsin) on the degree of hydrolysis: (**A**) Muga pupae whole protein concentrate (MPPC) and (**B**) Eri pupae whole protein concentrate (EPPC).

**Figure 2 nutrients-15-01035-f002:**
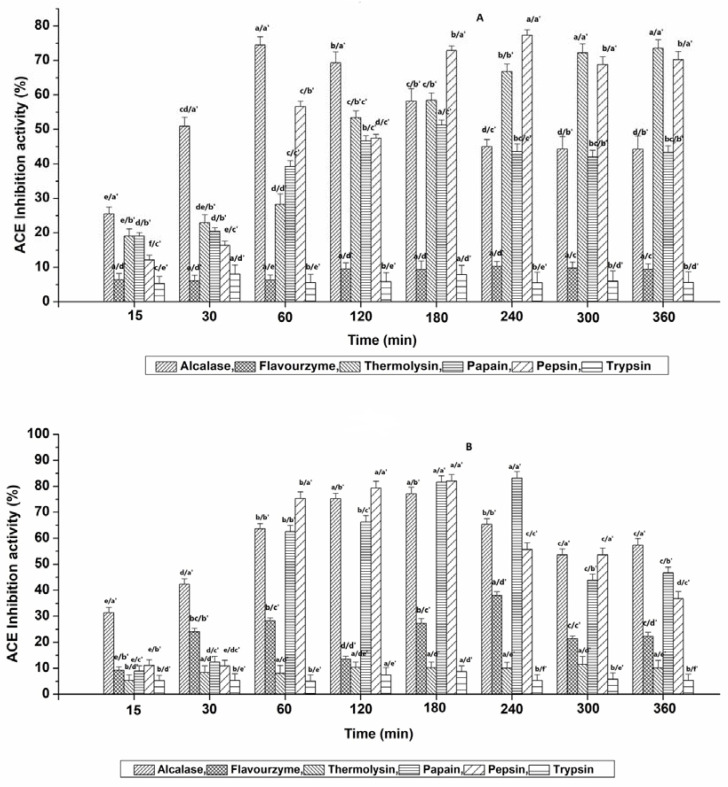
Influence of six proteases on the ACE-inhibition activity of hydrolysates obtained from (**A**) Muga pupae protein extract (MPPC) and (**B**) Eri pupae protein extract (EPPC). The values are expressed as mean ± SD of triplicate tests. Statistical difference (*p* < 0.05) within the same enzyme at different time points is denoted by letters a–e, and difference within six various enzymes at each time point is denoted with letters a′–e′.

**Figure 3 nutrients-15-01035-f003:**
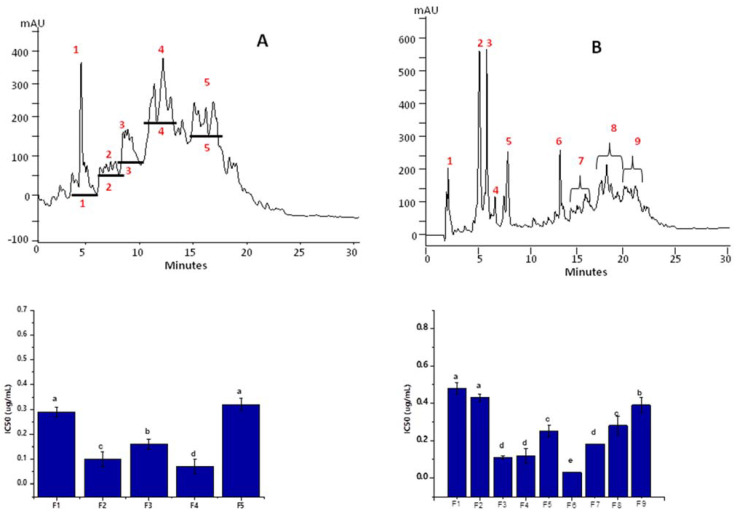
Elution profiles of Alc_M60min_F3 and Pap_E240min_F3 fractions by RP-HPLC and the ACE-inhibitory activity of each sub-fraction; (**A**) Alc_M60min_F3, (**B**) Pap_E240min_F3. Red numbers in the chromatograms indicate the peptides’ sub-fractions F1-F5 of Alc_M60min_F3 and F1-F9 of Pap_E240min_F3. Statistical difference (*p* < 0.05) is denoted by letters a–c. Shared letters indicate no significant difference.

**Figure 4 nutrients-15-01035-f004:**
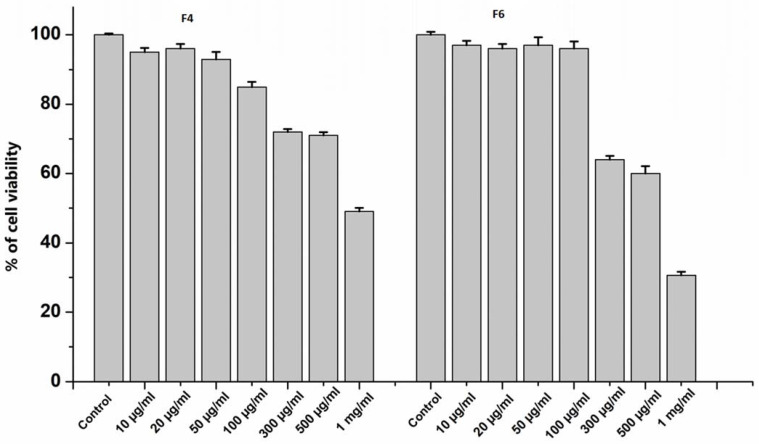
Cell viability of HUVECs treated with F4 and F6 at different concentrations (0, 10, 20, 50, 100, 300, 500 and 1000 µg mL^−1^) for 24 h. Data are the results of the averages of at least three different experiments.

**Figure 5 nutrients-15-01035-f005:**
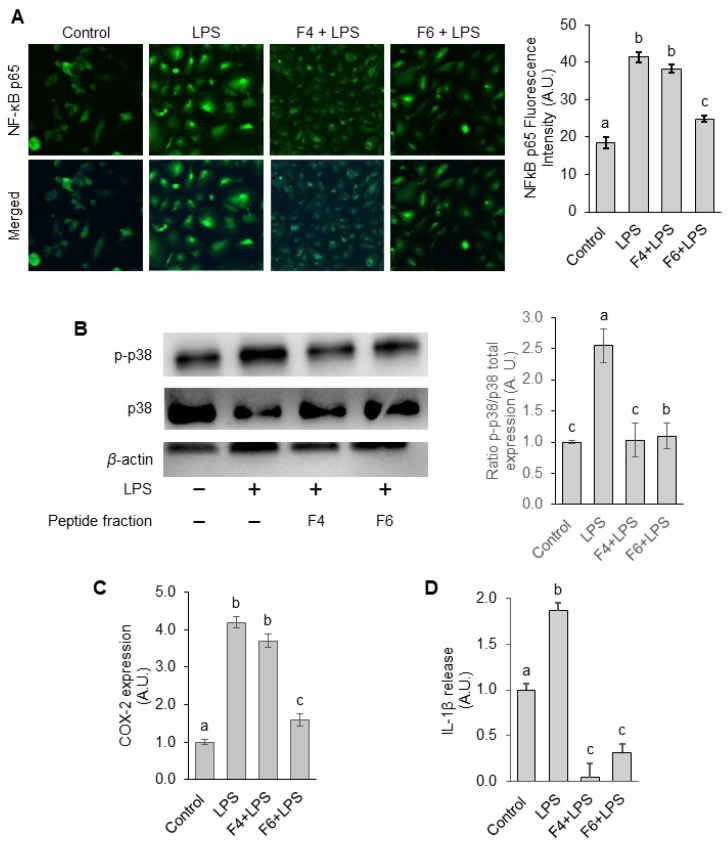
Effects of F4 and F6 on LPS-induced inflammatory processes in endothelial cells. (**A**) NF-κB p65 nuclear translocation was evaluated by confocal microscopy after 3 h of LPS stimulation. Images were captured and p65 nuclear translocation quantified using ImageJ software. (**B**) Phosphorylation of p38 MAPK was measured by immunoblotting. Protein bands of phospho-p38 were normalized to the total form and the values in the groups were relative values to the control group. Protein bands were quantified by densitometry and normalized to their respective loading controls. (**C**,**D**) COX-2 protein expression and IL-1β secretion at 12 h of LPS stimulation were assessed by ELISA assay. The protein level in the control group was set as 1 and the values in the other groups were relative values to the control group. Statistical difference (*p* < 0.05) is denoted by letters a–c. Shared letters indicate no significant difference.

**Figure 6 nutrients-15-01035-f006:**
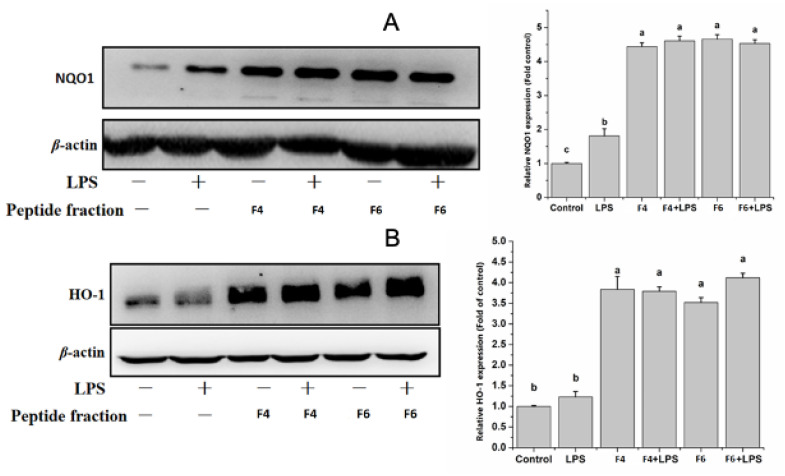
(**A**,**B**) Effects of F4 and F6 on LPS-induced antioxidant responses in endothelial cells. Eighteen hours after LPS addition, HO-1 and NQO-1 expression was measured by immunoblotting. The contents in the control groups were set as 1, and the values in the other groups were relative values to the control group. Protein bands were quantified by densitometry and normalized to their respective loading controls. The values are expressed as mean ± SD. Data are the results of the averages of at least three different experiments. Statistical difference (*p* < 0.05) is denoted by letters a–c. Shared letters indicate no significant difference.

**Table 1 nutrients-15-01035-t001:** ACE inhibition and antioxidant activities of selected hydrolysates.

S. No.	Hydrolysates	ACE Inhibition (%)	ACE Inhibition (IC_50_ mg mL^−1^)	DPPH (EC_50_ mg mL^−1^)	ABTS (EC_50_ mg mL^−1^)
1.	Alc_M60min	74.46 ± 2.44	0.17 ± 0.02 ^f^	1.01 ± 0.05 ^c^	2.45 ± 0.48 ^c^
2.	Ther_M360min	73.59 ± 2.39	1.52 ± 0.14 ^a^	2.58 ± 0.14 ^ab^	2.34 ± 0.19 ^c^
3.	Pep_M180min	72.89 ± 1.36	1.29 ± 0.21 ^ab^	0.9 ± 0.13 ^c^	4.02 ± 1.04 ^a^
4.	Pep_M240min	77.35 ± 1.57	0.12 ± 0.01 ^f^	2.08 ± 0.38 ^b^	3.47 ± 1.05 ^b^
5.	Alc_E180min	77.05 ± 2.12	0.65 ± 0.21 ^d^	0.75 ± 0.21 ^c^	3.52 ± 1.11 ^b^
6.	Pap_E240min	83.21 ± 2.51	0.33 ± 0.11 ^e^	0.48 ± 0.27 ^d^	2.08 ± 0.64 ^cd^
7.	Pep_E60min	75.31 ± 2.57	1.71 ± 0.29 ^a^	3.88 ± 0.91 ^a^	1.26 ± 0.27 ^d^
8.	Pep_E180min	82.07 ± 2.56	0.87 ± 0.18 ^bc^	2.11 ± 0.57 ^b^	3.41 ± 0.28 ^b^

All the values are expressed as mean ± SD of duplicate tests. Shared letters in the same column indicate no significant difference (*p* < 0.05). Alc_M60min, Alcalase-treated Muga pupae protein hydrolysate at 60 min; Ther_M360min, Thermolysin-treated Muga pupae protein hydrolysate at 360 min; Pep_M180min, Pepsin-treated Muga pupae protein hydrolysate at 180 min; Pep_M240min, Pepsin-treated Muga pupae protein hydrolysate at 240 min; Alc_E180min, Alcalase-treated Eri pupae protein hydrolysate at 180 min; Pap_E240min, Papain-treated Eri pupae protein hydrolysate at 240 min; Pep_E60min, Pepsin-treated Eri pupae protein hydrolysate at 60 min; Pep_E180min, Pepsin-treated Eri pupae protein hydrolysate at 180 min.

**Table 2 nutrients-15-01035-t002:** ACE inhibition and antioxidant activities of Alc_M60min and Pap_E240min hydrolysates’ ultrafiltrate fractions.

Fraction	Molecular Weight	ACE Inhibition (IC_50_ µg mL^−1^)	DPPH (EC_50_ µg mL^−1^)	ABTS (EC_50_ µg mL^−1^)
Alc_M60min_F1	MW > 10 kDa	16.57 ± 1.79 ^a^	144.13 ± 4.66 ^a^	70.73 ± 2.38 ^b^
Alc_M60min_F2	3 < MW < 10 KDa	2.34 ± 1.08 ^b^	129.58 ± 4.05 ^b^	56.35 ± 3.11 ^c^
Alc_M60min_F3	MW < 3 kDa	0.49 ± 0.25 ^c^	90.21 ± 3.27 ^c^	85.14 ± 4.63 ^a^
Pap_E240min_F1	MW > 10 kDa	56.45 ± 1.38 ^a^	66.41 ± 3.05 ^a^	88.76 ± 4.80 ^a^
Pap_E240min_F2	3 < MW < 10 KDa	10.6 ± 2.77 ^b^	58.21 ± 2.56 ^b^	23.71 ± 3.57 ^b^
Pap_E240min_F3	MW < 3 kDa	0.67 ± 0.16 ^c^	49.32 ± 4.09 ^c^	19.21 ± 3.64 ^b^

All the values are expressed as mean ± SD of duplicate tests. Shared letters in the same column indicate no significant difference (*p* < 0.05).

## Data Availability

The data presented in this study are available on request from the corresponding author.

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
