# Peer review of "Evaluation of Anti-Oxinflammatory and ACE-Inhibitory Properties of Protein Hydrolysates Obtained from Edible Non-Mulberry Silkworm Pupae *(Antheraea assama* and *Philosomia ricinii*)"

_nutrients, 2023, doi:10.3390/nu15041035_

Round 1
Reviewer 1 Report
Totally speaking, this menuscript regarding to the isolation and identification of biological acitve peptides obtained from edible insect-protein with grate potential to target complex pathological mechanisms responsible for chronic health conditions such as hypertension development. The main goal was to create novel food-derived bioactive peptides that would complement state-of-the-art research research on this topic. The non-mulberry silkworm pupae may offer a decent amount of protein and fats, according to the information from the current literature. The bioactive properties of the novel peptides with strong angiotensin-converting enzyme (ACE) inhibitory activity were discussed in this manuscript. The dead-end ultrafiltration process was used for the preparation of the peptide fractions, which showed the highest antioxidant and ACE-inhibitory activities. The reported and discussed results showed that purified peptide subfractions showed highest ACE-inhibition as well as potent anti-oxinflammatory activities in lipopolysaccharide (LPS)-treated endothelial cells. However, there are a few points that must be addressed.
(1) Abstract section
• "In this study, enzymatic protein hydrolysates from non-mulberry edible silkworm Antheraea assama (Muga) and Philosomia ricini (Eri) pupae, specifically alcalase (A. assama) and papain (P. ricini) hydrolysates obtained after 60 and 240 min, exhibited highest ACE inhibitory and antioxidant properties." Please describe the prepared hydrolysates in a manner that is more scientifically acceptable, i.e., start by stating that you used food-grade proteases for hydrolysis and that the hydrolysis was completed between 60 and 240 minutes, then list your substrates and assign the prepared hydrolysates the corresponding abbreviations.
• "Alc_M60min_F3 and Pap_E240min_F3" Abbreviations cannot be used in this chapter of the paper without explanation. Here, the only thing that is clear is that there is an F3 fraction under serial number three with peptides of molecular masses within a defined range, but many readers may get confused. It's a very confusing part of the Alc_M60 abbreviation. Although it is highly unclear to the reader, I can easily conclude that it is Alcalase from the substrate M for 60 minutes because I am an expert on the subject of peptide production by the action of proteolytic enzymes. Throughout the entire manuscript, fix this.
• "ultrafiltration" I suggest that the term "dead-end" ultrafiltration be used throughout the paper because, from a technological point of view, ultrafiltration procedures differ according to the flow of the feed (dead-end and cross-flow). I believe that it is important for the readers to be clear at the very beginning what the procedure is about, and that would avoid the parallel with the currently very popular application of ultrafiltration filters, which are used for concentrating samples.
• "sub-fractions" Write without dashes, only like subfractions.
(2) Introduction section
• Please expand the paragraph in the section titled "Introduction" that mentions diseases and biologically active food ingredients that can be utilized to lessen the symptoms of diseases, peptides in particular holding a specific place among these. Use up-to-date literature research. Indeed, highlight the importance of enzymatic hydrolysis, purification procedures, and separation of biologically active fractions using membrane separation processes and chromatography, and review the literature to highlight peptides of significant molecular weight for the biological properties examined in the manuscript. In the final paragraph of
Introduction section, describe the study's objectives in more detail. Save the final sentence for the conclusion or the abstract; this is where the surplus is.
(2) Materials and methods
• Alcalase, papain, pepsin, thermolysin, and Flavourzyme are listed as commercial enzyme preparations, but you don't specify which specific enzymes they are. Please describe this enzyme's kind (endo or exopeptidase). I believe it is crucial for readers to understand what kind of enzyme it is in addition to the brand name.
• The commercial name, model, manufacturer, and country of origin of each device used must be listed.
• "room temperature" Please, specify the exact temperature, whether it was 20 or 25 degrees.
• Why E/S ratio was 0.5%? Is that was mass ratio? Place, explain, and specify in the method where the E/S ratio was optimized, or cite the reference data.
• Why was the protein concentration set at 5%? I believe that it is more adequate to determine the E/S ratio through proteins because the substrate for proteases is proteins. So, when you calculate the E/S ratio, you measure the mass of enzyme in relation to the mass of protein in the reaction vessel, so it is better to calculate like this in your research; it will be clearer and easier to compare with the literature. Someone else can have a substrate that has only 30-40% proteins, and yours has 60%, and if they do an experiment like you with 0.5% E/S ratio, they can have better results just because there is much less proteins.
• Not just the E/S ratio needs to be specified; the substrate concentration during the enzymatic hydrolysis also.
• "digestions" Please substitute enzymatic hydrolysis or proteolysis for this term.
• Please write the equation for the calculation of the degree of hydrolysis and explain which value was used for the detrmination of htot for both substrates.
• "Peptide fractionation using ultrafiltration" In your research, centrifugal filter units were used. Please, explain how much time you performed cycle for one molecular weight? What was the separation speed and time duration? Why didn't you use an ultrafiltration cells or membrane modules, in which the processes of distributing molecules according to molecular weight and in accordance with the pore sizes of the membrane (cut-off), take longer, are more reliable, and certainly do not result in the solution passing and the particles being left behind on the membrane?
• "Hundred mg mL-1" Is the question about mass or concentration? It is not written chemically correctly. Write the concentration in numbers, not words, and indicate that it is the stock solution.
• "Hundred μL" Clarify this term. Also, rewrite the mobile phase that was used for the preparative chomatoraphy.
(3) All diagrams in the manuscript should be modified and increased in resolution in accordance with the journal's guidelines.
(4) There is a lack of a comparison between the achieved results and the data from the literature. The conclusion of the results' discussion is required.
(5) Please clarify the all aforementioned questions or suggestions and make the necessary revisions to the revised manuscript.
(6) It is advised that the authors recheck the main text during the revision to make this manuscript more readable.
(7) Correlation and regression analysis of degree of hydrolysis with ACE-inhibition activities obtained with supstrate-influenced hydrolysates:
• Correlation is not linear in all cases. It is necessary to determine the matematical model of the correlation; on measurements where the correlation is not linear, find what it will be, probably a second-order polynomial, and fit the experimental data with such a model, and extract the correlation coefficient from it. R2 must be greater than 0.9 in order for the results to be acceptable for publication. If the analysis is wrong, incorrect, it is best not to publish the currently displayed results as part of the supplementary data, but just remove them.
Author Response
Reviewer 1
Comments and Suggestions for Authors
The manuscript addresses the characterization of the anti-oxinflammatory and ACE-inhibitory properties of protein hydrolysates obtained from non-mulberry silkworm pupae (Antheraea assama and Philosomia ricinii), targeted to be explored as nutritional approaches against hypertension and related cardiovascular diseases.
The overall quality of the work is acceptable and the subject is relevant particularly in face of the new challenges regarding the need for an integral use of all the available biomass in an increasingly required circular economy.
The abstract is concise and gives the required information of the content of the article.
The Introduction addresses sufficiently well the state of the art and provides adequate justification for the research preformed.
Material and Methods gives, in general, all the information for repetition of the experiments.
- Change the symbol of degree from ºC to oC.
R. We are thankful to the reviewer for pointing out the error. The symbol of degree has been changed from ºC to oCthroughout in the revised manuscript.
- Taking into consideration the existence of possible seasonal variations in pupae composition (Bardoloi and Hazarika, 1992; https://doi.org/10.1093/ee/21.6.1398) the month of sample collection should be indicated.
R. We appreciate the suggestion of the reviewer and thankful for pointing out the error. The improvement in the revised manuscript is summarised below,
In line 103, page 3 of the revised manuscript, the sentence has been revised, which read as,
district, and some pupae were also procured in the month of march, 2017 from……………
- How is guaranteed the representativeness of the raw material extraction? How many extractions were made?
R. We understand the concern of the reviewer. Please see Line 139 onwards, page 3 in the revised manuscript which reads as,
“Supernatants obtained from total four extractions were adjusted to pH 4.0 - 5.8, using 1 M HCl to screen the isoelectric point.”
- In M&M, section 2.12-Statistical analysis it is referred that data are indicated as mean ± SEM (Standard Error of the Mean), but if the legend of Figures 2 and 5 and in Tables 1 and 2 values are expressed as mean ± SD (Standard Deviation). Considering that SEM gives an idea of the accuracy of the mean, and the SD gives an idea of the variability of single observations, please clarify the apparent inconsistency.
R. In the revised manuscript, we have highlighted in section 2.12 that all the results are expressed as mean ± Standard Deviation. Additionally, we have carefully checked the entire manuscript so as to not reflect such formatting error.
Presentation and discussion of results is done correctly and, in general, with adequate comparison to published information to support conclusions. The scientific quality of the work in this respect is good and the conclusions are supported by the results.
R. We thank the reviewer for the positive comments.
Other points of remark:
- Page 7
3.2. ACE-inhibitory potential of hydrolysates
Figure 1S. A represents Eri results and B Muga results. Change this and keep the logic of previous figures in terms of the attribution of A to Muga graphic results and B to Eri.
R. In section 3.2., representation of figure A and B has been revised accordingly as suggested by the reviewer in the revised manuscript.
- Explain better: “It was noted that with extending hydrolysis time, the ACE inhibition activity exhibited by alcalase-, papain- and pepsin-treated hydrolysates were mostly unstable for both types (supplementary data, Fig. 1S A and 1S B), despite of moderately increasing DH with time.”
R. We understand the concern of the reviewer. Please see Line 338 onwards, page 7 in the revised manuscript which reads as,
“This was probably because with continuous cleavage process, the new formed peptides may have either a strong affinity for ACE functional active site or may be further cleaved over time, resulting in production of large number of free amino acids (FAA) and more or less active peptides, as suggested by Salampessy et al. [22].”
- How the degree of hydrolysis compares with earlier studies carried out for other insects?
R. We have added the relevant references in the revised manuscript. Please see Line 317 onwards, page 7 in the revised manuscript which reads as,
“The DH value seen in this study for Alcalase hydrolysate (Muga and Eri) after 2 h was potentially greater than the published data shown by Dion Poulin et al. [18] for G. sigillatus and T. molitor (28.1 to 33.8% at 3.0% (E/S)) and Purschke et al. [19] for Locusta migratoria (13.3 - 15.2%; E/S = 1.0%). The results were consistent with data (29.2% - 51.2%; at 0.5%, 1.5% and 3.0% E/S ratio) studied by Hall et al. [12, 13] after 30 – 90 min for the same insects at 50% (w/v) concentration. Such variation might be influenced by prior heat treatment at 90 oC that can affect DH. Compared to DH results achieved for A. diaperinus (20.5% (0.5% E/S), 20.3% (1.5% E/S) and 25.0 - 28.4% (3.0% E/S); 4 h) using Alcalase as presented by Sousa et al. [20], these values were also considerably higher. DH values obtained for Flavourzyme derived hydrolysates for Muga pupae was found higher than reported data of cricket protein (33.0%; 2.0% E/S) but in DH noted for Eri pupae was similar to the values presented for Mealworm protein (51.0%) at similar conditions [21]. These differences can be explained by various factors like protein structure, amino acid sequence and other treatment conditions.”
8. Page 8
What is the benchmark reference for the statement?
3.3. Antioxidative potential of selected hydrolysates
“Based on our data, it can be confirmed that the selected hydrolysates also efficiently displayed antioxidant capacities.”
R. Please see Line 405 onwards, page 9 in the revised manuscript which reads as,
Although the strongest antiradical property was very lower than the IC50 value obtained for raw (5.3 µg/mL), boiled form of T. molitor (28.9 µg/mL); baked G. sigillatus and S. gregaria (28.5 µg/mL) [25]. These values were also lesser than the ones obtained for Zophobas morio (4.6 µg/mL) against ABTS, Amphiacusta annulipes (19.1 µg/mL) and silkworm protein (57.91 µg/mL) hydrolysate against DPPH achieved after in vitro gastrointestinal digestion [26, 27].
Presentation of results and discussion of data presented in Table 1 regarding Fe2+ chelating activity and Reducing power has not been done.
R. We thank the reviewer for noticing the mistake, indeed we have modified the table accordingly to the presented results.
3.4. Antioxidative and ACE inhibition potential of ultrafiltrates
“When ACE inhibitory and antioxidative properties of eight selected hydrolysates were analyzed, it appeared that the hydrolysate Alc_M60min and Pap_E240min presented the most efficient bioactivities relative to others.”
Be more exact/specific than “…it appeared…”
R. We have modified the sentence accordingly.
3.6. Anti-oxinflammatory activity of purified peptide fractions F4 and F6 in human endothelial cells
“…generating a vicious cycle of oxinflammation…”
R. The correct expression is vicious circle.
R. We have corrected the sentence as suggested.
It is suggested the division of Figure 5 in 2 or 3 figures in order to improve its readability. Also, the quality of the photos and the text in the axis needs improvement. The methodology described in the legend of Figure 5 should be in the appropriate section in Material and Methods.
R. We have corrected the figure and the legends as suggested. Indeed Figure 5 was split in figure 5 (A-D) and Figure B (A-B).

Reviewer 2 Report
The manuscript addresses the characterization of the anti-oxinflammatory and ACE-inhibitory properties of protein hydrolysates obtained from non-mulberry silkworm pupae (Antheraea assama and Philosomia ricinii), targeted to be explored as nutritional approaches against hypertension and related cardiovascular diseases.
The overall quality of the work is acceptable and the subject is relevant particularly in face of the new challenges regarding the need for an integral use of all the available biomass in an increasingly required circular economy.
The abstract is concise and gives the required information of the content of the article.
The Introduction addresses sufficiently well the state of the art and provides adequate justification for the research preformed.
Material and Methods gives, in general, all the information for repetition of the experiments.
Change the symbol of degree from ºC to oC.
Taking into consideration the existence of possible seasonal variations in pupae composition (Bardoloi and Hazarika, 1992; https://doi.org/10.1093/ee/21.6.1398) the month of sample collection should be indicated.
How is guaranteed the representativeness of the raw material extraction? How many extractions were made?
In M&M, section 2.12-Statistical analysis it is referred that data are indicated as mean ± SEM (Standard Error of the Mean), but if the legend of Figures 2 and 5 and in Tables 1 and 2 values are expressed as mean ± SD (Standard Deviation). Considering that SEM gives an idea of the accuracy of the mean, and the SD gives an idea of the variability of single observations, please clarify the apparent inconsistency.
Presentation and discussion of results is done correctly and, in general, with adequate comparison to published information to support conclusions. The scientific quality of the work in this respect is good and the conclusions are supported by the results.
Other points of remark:
Page 7
3.2. ACE-inhibitory potential of hydrolysates
Figure 1S. A represents Eri results and B Muga results. Change this and keep the logic of previous figures in terms of the attribution of A to Muga graphic results and B to Eri.
Explain better: “It was noted that with extending hydrolysis time, the ACE inhibition activity exhibited by alcalase-, papain- and pepsin-treated hydrolysates were mostly unstable for both types (supplementary data, Fig. 1S A and 1S B), despite of moderately increasing DH with time.”
How the degree of hydrolysis compares with earlier studies carried out for other insects?
Page 8
What is the benchmark reference for the statement?
3.3. Antioxidative potential of selected hydrolysates
“Based on our data, it can be confirmed that the selected hydrolysates also efficiently displayed antioxidant capacities.”
Presentation of results and discussion of data presented in Table 1 regarding Fe2+ chelating
activity and Reducing power has not been done.
3.4. Antioxidative and ACE inhibition potential of ultrafiltrates
“When ACE inhibitory and antioxidative properties of eight selected hydrolysates were analyzed, it appeared that the hydrolysate Alc_M60min and Pap_E240min presented the most efficient bioactivities relative to others.”
Be more exact/specific than “…it appeared…”
3.6. Anti-oxinflammatory activity of purified peptide fractions F4 and F6 in human endothelial cells
“…generating a vicious cycle of oxinflammation…”
The correct expression is vicious circle.
It is suggested the division of Figure 5 in 2 or 3 figures in order to improve its readability.
Also, the quality of the photos and the text in the axis needs improvement.
The methodology described in the legend of Figure 5 should be in the appropriate section in Material and Methods.
Author Response
Reviewer 2
Comments and Suggestions for Authors
A very interesting and well written, well thought out work illustrating the efficient recovery of underutilised proteins from non-mulberry silkworm waste after the reeling process and the preparation of their hydrolysates, which have interesting bioactivities. The authors have shown that enzyme-specific hydrolysis is a very important step in validating the production of multifunctional peptides that could have potential benefits for human health.
The methodology is very well described, so it is easy for other scientists to replicate the research. What I can not say about the Results and Discussion section.
I am dissatisfied with the analysis of the results obtained in comparison with those already published by other authors. Such interesting results should be discussed in many scientific sources.
I also miss graphic representations, an attempt to present the most important results in the form of a drawing that attracts potential people interested in the research.
I did not notice any glaring stylistic or linguistic errors in the manuscript. However, I would rewrite the introduction and the results and discussion sections. I would focus on simplifying the language to make it more accessible to people who do not study the topics described in depth.
The above information is only a guide for a better change of the manuscript, it does not diminish in any way the rank of the research and the written article, which in my opinion should be published after slight corrections.
R. We understand the concern of the reviewer to write the paper in a more layman way, however considering that it is a scientific journal we wanted to be as scientific as possible to make the science solid and convincing. As the reviewer will agree it is not easy to find a style that will please all the reviewers. We thank the reviewer for the suggestion and we hope that the changes made in the manuscript have improved its reading.
Reviewer 3 Report
A very interesting and well written, well thought out work illustrating the efficient recovery of underutilised proteins from non-mulberry silkworm waste after the reeling process and the preparation of their hydrolysates, which have interesting bioactivities. The authors have shown that enzyme-specific hydrolysis is a very important step in validating the production of multifunctional peptides that could have potential benefits for human health.
The methodology is very well described, so it is easy for other scientists to replicate the research. What I can not say about the Results and Discussion section. I am dissatisfied with the analysis of the results obtained in comparison with those already published by other authors. Such interesting results should be discussed in many scientific sources. I also miss graphic representations, an attempt to present the most important results in the form of a drawing that attracts potential people interested in the research. I did not notice any glaring stylistic or linguistic errors in the manuscript. However, I would rewrite the introduction and the results and discussion sections. I would focus on simplifying the language to make it more accessible to people who do not study the topics described in depth. The above information is only a guide for a better change of the manuscript, it does not diminish in any way the rank of the research and the written article, which in my opinion should be published after slight corrections.
Author Response
We thank the reviewer for the positive comments. The manuscript have checked by a native English speaker and typos have been corrected.